# A cluster randomised trial of a Needs Assessment Tool for adult Cancer patients and their carers (NAT-C) in primary care: A feasibility study

Joseph Clark[1]*, Elvis Amoakwa[1], Alexandra Wright-Hughes[2], John Blenkinsopp[3], David C. Currow[1], David Meads[4], Amanda Farrin[2], Victoria Allgar[5], Una Macleod[6], Miriam Johnson[1]

1 Wolfson Palliative Care Research Centre, University of Hull, Hull, United Kingdom, 2 Leeds Institute of Clinical Trials Research, University of Leeds, Leeds, United Kingdom, 3 University of Northumbria, Newcastle upon Tyne, United Kingdom, 4 Leeds Institute of Health Sciences, University of Leeds, Leeds, United Kingdom, 5 Hull York Medical School, University of York, York, United Kingdom, 6 Hull York Medical School, University of Hull, Hull, United Kingdom

* joseph.clark@hyms.ac.uk

**Data Availability Statement:** Our ethics approved consent process does not allow access unless there has been a process to identify that the person

## Abstract

### Background

People with cancer often have unidentified symptoms and social care needs. The Needs Assessment Tool-Cancer (NAT-C) is a validated, structured method of assessing patient/carer concerns and prompting action, to address unmet need.

### Aims

Assess feasibility and acceptability of a definitive two-armed cluster randomised trial of NAT-C in primary care by evaluating: recruitment of GP practices, patients and carers; most effective approach of ensuring NAT-C appointments, acceptability of study measures and follow-up.

### Methods

Non-blinded, feasibility study in four General Practices, with cluster randomisation to method of NAT-C appointment delivery, and process evaluation. Adults with active cancer were invited to participate with or without carer. Practices cluster randomised (1:1) to Arm I: promotion and use of NAT-C with a NAT-C trained clinician or Arm II: clinician of choice irrespective of training status. Participants completed study questionnaires at: baseline, 1, 3 and 6 months. Patients booked a 20 minute needs-assessment appointment post-baseline. Patients, carers and GP practice staff views regarding the study sought through interviews/ focus groups. Quantitative data were analysed descriptively. Qualitative data were analysed thematically, informed by Normalisation Process Theory. Progression to a definitive trial was assessed against feasibility outcomes, relating to: recruitment rate, uptake and delivery of the NAT-C, data collection and quality.

accessing the data is a researcher. Requests for the datasets used and/or analysed during the current study are available on request from Yorkshire & The Humber - Leeds East Research Ethics Committee, Jarrow Business Centre, Rolling Mill Road, Jarrow, NE32 3DT. Telephone: 0207 104 8081.

**Funding:** This study was funded by Yorkshire Cancer Research (Registered Charity 516898), grant award number H397. MJ was the Principal Applicant. The Co-Applicants were UM, AJF, DM, AWH, JB, DC, RF, SW, VLA, HSFF. No commercial companies funded either the study or any of the authors. The study was sponsored by the University of Hull (www.hull.ac.uk). The funders had no role in study design, data collection and analysis, decision to publish, or preparation of the manuscript.

**Competing interests:** The authors have declared that no competing interests exist.

## Results

Five GP practices approached, four recruited and trained to use the NAT-C. Forty-seven participants and 17 carers recruited. At baseline, 34/47 (72%) participants reported at least one moderate-severe unmet need, confirming study rationale. 32/47 (68%) participants received a NAT-C-guided consultation, 19 of which on Arm I. Study attrition at one month (n = 44 (94%), n = 16 (94%)), three months (n = 38 (81%), n = 14 (82%)) and six months (n = 32 (68%), n = 10 (59%)). Fifteen patient interviews conducted across the whole study and one focus group *at each* GP practice. Participants supported a definitive study and found measures acceptable.

## Conclusion

The feasibility trial indicated that recruitment rate, intervention uptake and data collection were appropriate, with refinements, for a definitive multi-centre cluster randomised controlled trial. Feasibility outcomes informed the design of a 2-armed cluster randomised controlled trial to test the effectiveness and cost-effectiveness of the NAT-C compared with usual care.

## Introduction

Many people with cancer and their carers have physical, psychological, emotional and social problems, which commonly go unidentified or addressed. Unmet needs in patients with cancer and their carers are common but poorly identified and addressed [1]. One consequence of this, is that access to palliative care and other supportive services is often based upon prognosis, rather than present need and that many people who might benefit from such services, do not have access to them [2]. Early identification of patient and carer problems in primary care may help improve access to palliative care and other supportive services and improve the quality of life of cancer patients and their carers.

One way in which primary care is responsive to the needs of cancer patients, is through the provision of a Cancer Care Review (CCR) as part of the Quality and Outcome Framework [3]. A CCR aims to help the person affected by cancer to understand what information and support is available to them locally. GP practices receive Quality and Outcomes Framework payments for conducting cancer reviews within six months of patients' diagnosis. However, there is no guidance regarding what a cancer review should involve or evidence-based template available. A clinician-led, systematic approach to symptom assessment may be more effective than general enquiry [4]. The United Kingdom (UK) National Cancer Action Team identified the need for tools to help holistic assessment [5] but few are developed and evaluated critically for use in primary care where practitioners are well placed to conduct such assessments [6].

A needs assessment tool provides a consistent and comprehensive system to prompt discussion of patients' support and care needs, helps triage tailored action to address patient concerns, and provides data for use in audit and service planning [7]. The Needs Assessment Tool–Cancer (NAT-C) is a structured *aide memoire* for more systematic patient and carer assessment. The NAT-C is a structured form, comprising five sections: Priority referral for further assessment (highlighting any need for palliative care referral), Patient wellbeing, Ability of carer or family to care for patient, Carer/Family wellbeing and Resulting referrals (if required). Where problems are identified, the tool encourages clinicians to report 'action taken'. The NAT-C was developed in Australia, and has been shown to reduce unmet informational needs of cancer

patients in the oncology clinic [7]. The tool has been adapted and psychometrically validated for use in UK primary care, but has not been tested for patient benefit in this setting [8].

Evidence-based cancer interventions are needed to improve holistic cancer care in primary care. However, delivery of clinical trials in primary care I known to be challenging, many struggle recruit to sample size and time-delays are common [9]. The NAT-C may reduce unmet supportive and palliative care needs of cancer patients and their carers by supporting clinicians to systematically assess patient/carer problems across multiple domains and take action. Identified problems may be managed in primary care or through referral to specialist services. In line with Medical Research Council (MRC) guidance, before conducting a Phase III (an MRC Framework for Complex Interventions evaluation phase trial [10]) to test the effectiveness of the NAT-C, we needed to address a number of uncertainties in relation to the feasibility of such. This report, therefore, presents the findings of a feasibility cluster Randomised Trial.

This trial was non-controlled (that is, all participating clusters received training in the NAT-C) but was randomised to allow comparison of two ways of delivering the NAT-C consultations. Our objectives were to evaluate: 1) feasibility of recruiting general practitioner (GP [family doctors]) practices; 2) feasibility of training clinicians to use the NAT-C; 3) method of ensuring delivery of a NAT-C appointment in relation to recruitment rate of patients (and carers) and practice preference; 4) proportion of patients unmet patient needs at baseline to investigate rationale for a definitive trial; 5) uptake of the NAT-C by clinicians; 6) completion rates of patient questionnaires; and 7) acceptability of study measures and processes.

## Materials and methods

### Study design

The Cancer Needs Assessment (CANASSESS) study was a non-blinded, parallel non-controlled, feasibility trial with cluster randomisation in primary care. A qualitative sub-study and process evaluation was also conducted in parallel to the feasibility trial. The study determined *a priori* stop/go criteria with regard to progressing to the planned Phase III trial. We report quantitative and qualitative feasibility findings with key aspects of the process evaluation.

### Participants and setting

We aimed to recruit four general practices within Hull, East Ridings and North Lincolnshire Clinical Commissioning Groups (CCGs). An Expression of Interest (EOI) request was administered to general practices in these regions by the Clinical Research Network (CRN), after which interested practices contacted the research team directly. We approached interested practices and entered discussions regarding participation in the order in which practices expressed interest. Eligible GP-practices were willing identify and invite eligible patients, to be trained to use the NAT-C and provide NAT-C-guided patient needs assessment consultations. Practice-level consent was provided by the Practice Manager.

**Randomisation.** Randomisation was to the method of delivery of the NAT-C appointment: patient attendance with a clinician they may not have met before, versus a known clinician; to test uncertainty regarding pragmatism versus optimising uptake of the NAT-C guided appointment. Practices were randomised to either Arm I) known NAT-C trained clinician (directed encounter), or; Arm II) any clinician irrespective of training status (undirected encounter), using 1:1 randomisation generated by using Randomization.com ⟨http://www.randomization.com⟩ by a senior statistician (VA). Practices were randomised following entry of all sites to the study and informed of their allocation by the trial manager.

**Patient / carer recruitment.** Eligible patients were initially consenting adults with incurable cancer, able to undertake study procedures. Patients living in an institutional care setting

were excluded. Patients were identified and invited by the clinical team using the practice's cancer register, or opportunistically during routine clinics. A site clinician screened returned registry screen results against the eligibility criteria.

Potential participants were informed invited *via* letter and provided with an information sheet, detailing the intervention and the purpose of the study. A 'flag' was placed on eligible patients' record, to indicate that they were eligible for the study, to aid opportunistic recruitment. Interested patients contacted the research team to express willingness to take part and arrange a baseline visit from a researcher who took informed consent prior to data collection.

Eligible carers were those nominated by the patient and able to provide informed consent. Carers were given an information sheet at the baseline visit and willing participants provided written informed consent prior to data collection.

After slow initial recruitment, inclusion criteria were amended to include patients with active cancer irrespective of treatment intent and allowed inclusion of a pre-paid Expression of Interest Form (EOI) with study invitations and patient information.

**Intervention.** All practice clinicians were invited to a face-to-face NAT-C training event provided by the study team at their own practice. Face-to-face training was delivered by a trial researcher and included: the rationale for the NAT-C, how to use the NAT-C and a real-life video-recorded patient NAT-C guided consultation to encourage clinicians to familiarise themselves with the NAT-C. Online training was also available with the same content. Clinicians could contact the research team with any questions.

Patients on each arm received a needs assessment appointment either in clinic or in patients' homes with a practice clinician (doctor or nurse). The NAT-C was available on paper, or electronically (EMIS, SystmOne). In arm I, only NAT-C trained clinicians were to conduct needs assessment. In arm II, participants could book an appointment with the clinician of their choice irrespective of training status. Clinicians were encouraged to conduct a needs assessment consultation 'as usual', but use the NAT-C as an *aide memoire* towards the end of an appointment, to ensure holistic assessment. Based on the original NAT-C work, assessment time was estimated at 20 minutes [8].

**Study assessments and outcomes.** Baseline participant and carer questionnaire completion occurred face-to-face with the researcher before delivery of the NAT-C. During the baseline visit, a researcher completed the AKPS and retrieved information on co-morbidities form patients' medical record to populate the CCI.

Patients completed a questionnaire booklet containing the Supportive Care Needs Survey SF34 (SCNS) and other questionnaires reported in S1 Box. Carer-participants completed a booklet containing the Carer Support Needs Assessment (CSNAT) and the Carer Experience Survey (CES). A researcher was present to explain the questionnaires and answer any questions from participants.

Follow up questionnaires at one, three and six months were collected according to participant choice (face-to-face, telephone, postal).

Completed NAT-C assessments were retrieved from the clinical record.

Feasibility outcomes, assessment measures and stop/go criteria are presented in S1 Box.

**Process evaluation.** *Normalisation MeAsure Development questionnaire (NoMAD) survey.* Ongoing feedback relating to feasibility/acceptability of study measures took place throughout the trial, mainly in the form of collating views of participants and GP practice staff, and researcher observation (JC/EA).

Following study training, clinicians were asked to complete an adapted NoMAD survey, informed by Normalisation Process Theory (NPT). The NoMAD instrument is a 23-item instrument for measuring implementation processes from the perspective of professionals directly involved in the work of implementing complex interventions in healthcare. We

adapted the NoMAD instrument in to a 17-point checklist to specifically address perceptions of the NAT-C relating to: (coherence); how clinicians engage with the NAT-C (cognitive participation); enact it (collective action); and appraise its effects (reflexive monitoring) [11].

*Interviews and focus groups.* All practice staff were invited to participate in a focus group at their general practice; those consenting provided a convenience sample. A purposive sample of patient-participants (with and without carer) were invited to a single interview at one of the study time-points (baseline, 1, 3, or 6 months). Participants provided written consent prior to interview. Patient semi-structured interviews were conducted in a place of patient-choice. Focus groups were conducted by two researchers and took place at GP practices using a topic guide and researchers took field-notes.

An interview guide was developed from the research group expertise [S1 File]. Patient interviews explored their: experiences of taking part in the study, views relating to the acceptability of study questionnaires and procedures. Focus groups elicited clinician experiences of using the NAT-C and overall views on practices' participation in the study, to identify potential refinements for a definitive trial.

All interviews and groups were audio-recorded and transcribed.

*Sample size.* Forty to 60 patient participants across four General Practices was considered sufficient to address the quantitative feasibility questions relating to sample size estimation and recruitment rates. As this was a feasibility study, a formal calculation was not appropriate as effectiveness was not being assessed [12]. Sample size recommendations for feasibility and pilot studies vary but numbers between 12 and 50 seem to be acceptable to address variation around baseline measures [13]. A sample size of 15 patients was considered sufficient for thematic saturation for this focussed area of qualitative inquiry. All GP practice staff were invited to focus groups at the conclusion of the study.

## Analysis

Quantitative data were managed and analysed descriptively using IBM Statistics SPSS 24.

For the proposed primary outcome measure, the SCNS-SF34, 34 questions were grouped into five domains of need: psychological; health system and information; physical and daily living; support and care and sexuality [14]. We report the proportion of participants with 'unmet need', defined as at least one 'moderate' or 'high' need. Changes in mean score within domains and contingency tables were used to show changes to patient outcomes over time. Where ≥50% of questions were answered within a domain, the mean response was used to impute missing item (s). Otherwise, individual domain scores were missing. All further patient reported outcome measures (PROMS) data are reported descriptively in accordance with published guidelines.

Intervention fidelity was assessed by reporting completed rates of NAT-C 'expected items' (unless irrelevant, for example, if a patient has no carer). Domains of clinician concern and referral data are reported descriptively. Where the NAT-C form itself was not found, but there was evidence in the patient record that the NAT-C had been used and this was confirmed by the clinician concerned, this was counted as "NAT-C conducted", otherwise, an absent NAT-C was reported as 'not known'.

Interviews and focus groups were transcribed and analysed thematically [15]. Three transcripts were independently coded by three researchers (JC, FR, MJ) and a coding framework developed. All transcripts were coded using the framework and analysed thematically using mind-maps. Codes were grouped to form descriptive themes, and further revised following discussion into analytic themes. Several themes were developed from the data. In this paper, we present a single theme relevant to feasibility outcomes. Preliminary qualitative findings were presented to patient, carer and clinician-participants at a public engagement event.

Attendees felt that the analysis broadly reflected their concerns and themes were refined following open discussions.

NoMAD survey responses were analysed descriptively. Using field notes, summary reports were written identifying key issues in relation to the feasibility of the project at six time-points to inform process evaluation. Key barriers to feasibility were reported and data-driven solutions identified.

The trial is reported consistent with relevant Consolidated Standards of Reporting Trials [CONSORT] statements [16] and the NAT-C is described in accordance with items of the Template for Intervention Description and Replication (TIDieR) checklist [17].

**Ethics approval and consent to participate.** The trial was approved by Yorkshire & The Humber—Leeds East Research Ethics Committee (ref: 17/YH/0141) and the Health Research Authority (HRA). Institutional permissions were obtained and the trial was registered on the ISRCTN registry (ISRCTN22325477: URL http://www.isrctn.com/ISRCTN22325477) prior to recruitment. All procedures performed in studies involving human participants were in accordance with the ethical standards of the institutional and/or national research committee and with the 1964 Helsinki declaration and its later amendments or comparable ethical standards. All participants provided informed written consent prior to any trial procedures. No deviations from the study protocol were reported by study sites, nor identified by the research team during monitoring.

## Results

The study took place between January 2017 and September 2018. Recruitment was between September 2017 and March 2018, follow-up ceased August 2018.

### Recruitment feasibility

**GP Practices.** Five GP practices were approached and four recruited [Fig 1] meeting our feasibility objective. One practice withdrew from negotiations due to potential perceived financial impact on time required.

**Practice characteristics.** Practice patient list size ranged from 7250 to 21,221. Dedicated GP research time was available in 3/4 practices and dedicated research nurse time in 2/4 practices, one in each arm. Patient retention including the number of participants followed up, withdrawing, and the timing and reasons for withdrawal is presented in the CONSORT chart.

**Patient and carer recruitment.** Forty seven patients and seven carers were recruited, 32 and 7 respectively completed the study. During September and October 2017, 0 patients were recruited (0 per month). Following an amendment which broadened the inclusion criteria to include *any* patient with active cancer regardless of prognosis and allow inclusion of EOI forms with study invitations, early recruitment problems were solved. Between November 2017 and the end of March 2018, 47 patients were recruited (11 patients a month). Twenty-one patients were recruited to Arm I and 26 to Arm II. Recruitment ceased once the stop/go criteria for recruitment rate was achieved.

**Participant characteristics.** Average age of patient-participants was 70 (Standard Deviation (SD): 12.4, range: 34 to 88) and 32 (68%) were men [Table 1]. All patient-participants were White British. Mean performance status (AKPS) was 76.0, SD 17.7; range 30 to 100). The most common diagnosis was prostate cancer (n = 15) and most patient-participants lived with a partner or family member (n = 33). Baseline demographic and clinical characteristics of patient-participants were well balanced between the two arms other than for comorbidity (CCI mean score 1.52 Arm 1; 0.73 Arm 2).

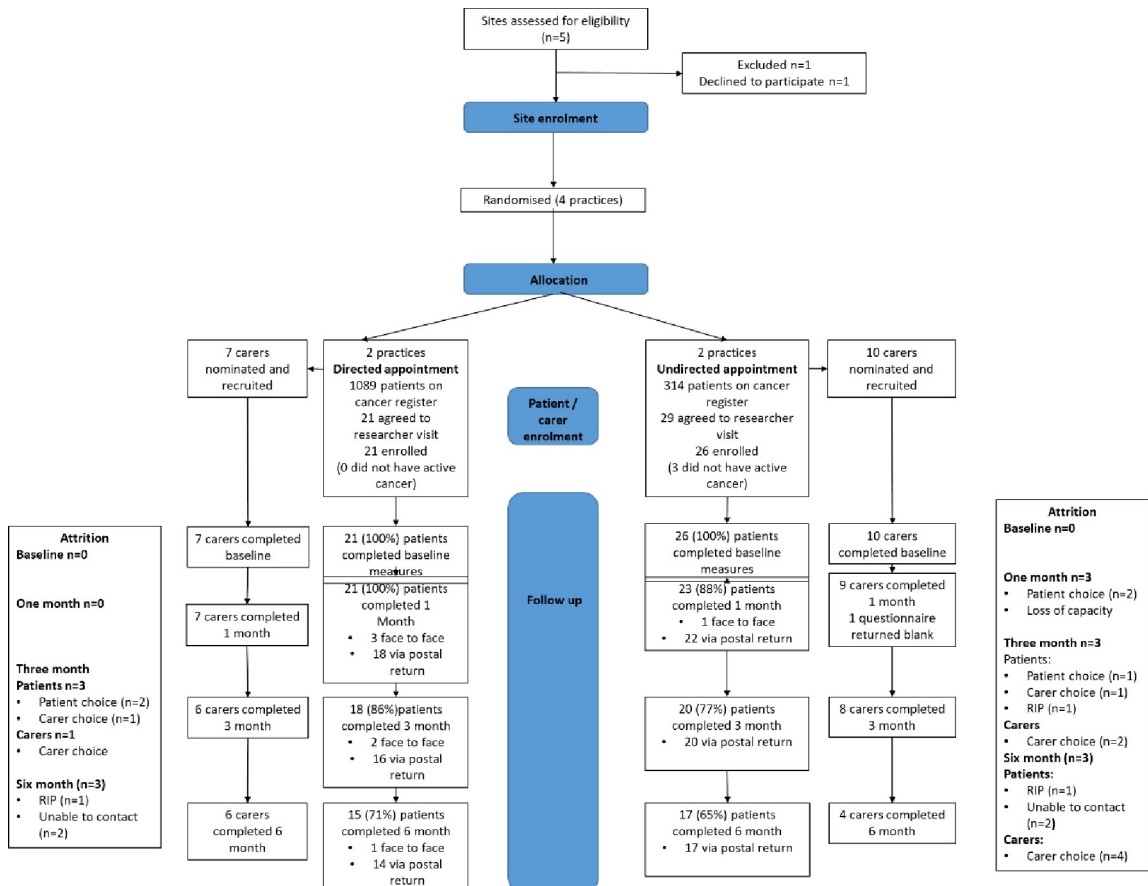

**Fig 1. CONSORT flow diagram: GP practice and patient/carer recruitment and attrition.**

**Intervention training, uptake and delivery.** NAT-C training of clinicians, scheduling and attendance at study appointments and delivery of the NAT-C, along with details of place of consultation, clinician conducting appointment, and length of appointment are presented in Table 2.

Of all practice staff, 24/33 (73%) GPs and 5/27 (19%) nurses were trained [Table 2], which met our feasibility objective regarding clinician willingness to be trained. The numbers of nurses relative to doctors trained varied widely across the practices. Most, 27/29 (93%), were trained face-to-face.

Of the 47 patient-participants recruited, 44 (94%) received a study appointment, of whom: 42/44 attended (96%), 38/42 (91%) received an appointment from a NAT-C-trained clinician and 32/42 (76%) received a NAT-C-guided consultation. More people received a NAT-C appointment when directed towards a known NAT-C clinician, informing trial design of a definitive study. Overall uptake and delivery of the NAT-C was reported for 32/47 (68%) patient-participants and demonstrated willingness of clinicians to use the NAT-C.

A completed NAT-C form could not be found for five patient-participants. NAT-C use was confirmed for two, but remained unknown for three. Three trained clinicians (2 GP, 1 nurse) conducted a study appointment without using the NAT-C and received further training.

The average time from participant recruitment to NAT-C appointment was 22 days (range 0, 66). Median length of NAT-C appointments was 15–20 minutes. In general, consultations in participants' home took longer and were completed on paper.

**Table 1. Patient and carer demographics.**

| | Patients | | | | |
|---|---|---|---|---|---|
| Demographics* | Directed encounter | | Undirected encounter | | Total |
| | Site 3 N = 15 | Site 4 N = 6 | Site 1 N = 12 | Site 2 N = 14 | N = 47 |
| **Sex** | | | | | |
| Male | 11 (73%) | 4 (67%) | 9 (75%) | 8 (57%) | 32 (68%) |
| Female | 4 (27%) | 2 (33%) | 3 (25%) | 6 (43%) | 15 (32%) |
| **Age** | | | | | |
| Mean (SD) | 71.6 (13.5) | 71.5 (13.0) | 71.6 (8.4) | 66.3 (14. 4) | 70.0 (12.4) |
| Median (Range) | 76 (34,88) | 75 (52,87) | 69 (57,87) | 70 (36,88) | 70 (34,88) |
| **Living arrangement and relationship status** | | | | | |
| Spouse/partner | 10 (67%) | 4 (67%) | 6 (50%) | 10 (71%) | 30 (64%) |
| Single/widowed/divorced | 4 (27%) | 2 (33%) | 5 (42%) | 3 (21%) | 14 (30%) |
| Other(Son/parents) | 1 (7%) | 0 (0%) | 1 (8%) | 1 (7%) | 3 (6%) |
| **Primary diagnosis** | | | | | |
| Prostate | 5 (33%) | 3 (50%) | 6 (50%) | 1 (7%) | 15 (32%) |
| Other | 4 (27%) | 0 (0%) | 5 (42%) | 4 (29%) | 13 (28%) |
| Breast | 0 (0%) | 1 (17%) | 0 (0%) | 3 (21%) | 4 (9%) |
| Lung | 2 (13%) | 0 (0%) | 0 (0%) | 1 (7%) | 3 (6%) |
| Hepatobiliary/Pancreatic | 0 (0%) | 0 (0%) | 0 (0%) | 3 (21%) | 3 (6%) |
| Urological (non-prostate) | 1 (7%) | 0 (0%) | 0 (0%) | 2 (14%) | 3 (6%) |
| Colorectal | 0 (0%) | 1 (17%) | 1 (8%) | 0 (0%) | 2 (4%) |
| Gynaecological | 1 (7%) | 1 (17%) | 0 (0%) | 0 (0%) | 2 (4%) |
| Brain | 1 (7%) | 0 (0%) | 0 (0%) | 0 (0%) | 1 (2%) |
| Skin | 1 (7%) | 0 (0%) | 0 (0%) | 0 (0%) | 1 (2%) |
| **Extent of cancer** | | | | | |
| Loco-regional | 11 (73%) | 4 (67%) | 11 (92%) | 12 (86%) | 38 (81%) |
| Metastatic | 4 (27%) | 2 (33%) | 1 (8%) | 2 (14%) | 9 (19%) |
| **Charlson Co-morbidity* Index score** | | | | | |
| Mean (Standard deviation) | 1.7 (2.3) | 1.2 (1.1) | 0.8 (0.9) | 0.6 (0.8) | 1.1 (1.6) |

**Fidelity to the intervention.** Item completion of the NAT-C was high; 93% completion across all items for 30 participants with data available, although 15 (50%) had at least one missing item. No pattern of non-completion was apparent.

The most common concerns were patient-participants' physical symptoms. No clinicians had concerns that beliefs, cultural or social factors were making care more complex. Four referrals resulted from consultations, one to specialist palliative care, two to social workers and one to a social prescriber (e.g. referrals to a community group). Some clinicians noted concerns, but did not complete 'action taken'.

**Baseline and follow up.** At baseline, 34/47 (72%) participants reported at least one moderate to severe unmet need confirming rationale for the study. Fig 1 shows proportions of patient-participants and carer-participants remaining in the study. Patient-participants withdrawn and lost to follow up generally had worse health outcomes across questionnaires at baseline; all patient-participants lost to follow-up at each time point had reported unmet need on the SCNS at baseline, compared to a baseline proportion of 34 (72%) [S2 File].

**Participant questionnaires.** There was high item completion for the proposed primary outcome measure, the SCNS-SF34 (99.9%, 98.1%, and 95.6% at baseline, 1, 3 and 6 months) supporting use of the SCNS as primary outcome in a definitive trial. The three least completed items across all time points were 'need for additional support with: 'changes in sexual feelings',

**Table 2. Study training and appointments.**

| | Directed encounter | | Undirected encounter | | Total |
|---|---|---|---|---|---|
| | Site 3 | Site 4 | Site 1 | Site 2 | |
| **Clinician training** | | | | | |
| **Total clinicians available** | 14 | 28 | 6 | 12 | 60 |
| GP | 8 (57%) | 15 (54%) | 3 (50%) | 9 (75%) | 33 (55%) |
| Nurse | 6 (43%) | 13 (46%) | 3 (50%) | 6 (25%) | 27 (45%) |
| **Total clinicians trained (% of total available)** | 4 (29%) | 10 (36%) | 6 (100%) | 9 (75%) | 29 (58%) |
| GP | 3 (38%) | 10 (67%) | 3 (100%) | 8 (89%) | 24 (73%) |
| Nurse | 1 (17%) | 0 (0%) | 3 (100%) | 1 (17%) | 5 (19%) |
| **Mode of training** | | | | | |
| Face to face | 4 (100%) | 10 (100%) | 4 (66.6%) | 9 (100%) | 27 (93%) |
| Online | 0 (0%) | 0 (0%) | 2 (33.3%) | 0 (0%) | 2 (7%) |
| **Timing of initial training in relation to first recruit** | | | | | |
| Days | 143 | 40 | 112 | 69 | |
| **Patient-participant study appointments** | | | | | |
| **Total patients recruited** | 15 | 6 | 12 | 14 | 47 |
| **Appointment scheduled** | | | | | |
| Yes | 15 (100%) | 5 (83.3%) | 11 (91.7%) | 13 (92.8%) | 44 (93.6%) |
| No[a] | 0 (0%) | 1 (16.7%) | 1 (8.3%) | 1 (7.2%) | 3 (6.3%) |
| **Patient attended appointment?** | | | | | |
| Yes | 15 (100%) | 5 (100%) | 10 (90.9%) | 12 (92.3%) | 42 (95.5%) |
| No[b] | 0 (0%) | 0 (0%) | 1 (9.1%) | 1 (7.7%) | 2 (4.5%) |
| **Time from randomisation to appointment (days)** | | | | | |
| Mean (SD) | 18 (12.4) | 22 (7.7) | 20 (17.6) | 28 (18.9) | 22 (15.7) |
| Median (range) | 14 (7–66) | 27 (14–29) | 11 (0–48) | 27 (7–59) | 20 (0–66) |
| **Location of appointment** | | | | | |
| GP practice | 9 (60%) | 5 (100%) | 9 (90%) | 12 (100%) | 35 (83.3%) |
| Patient home | 6 (40%) | 0 (0%) | 1 (10%) | 0 (0%) | 7 (16.7%) |
| **Clinician conducting appointment** | | | | | |
| GP | 6 (40%) | 5 (100%) | 5 (50%) | 12 (100%) | 27 (64.3%) |
| Nurse | 9 (60%) | 0 (0%) | 5 (50%) | 0 (0%) | 15 (35.7%) |
| **Uptake of NAT-C** | | | | | |
| **Appointment with a clinician trained in use of NAT-C?** | | | | | |
| Yes | 15 (100%) | 5 (100%) | 9 (90%) | 9 (75%) | 38 (90.5%) |
| No | 0 (0%) | 0 (0%) | 1 (10%) | 3 (25%) | 4 (9.5%) |
| **NAT-C used in appointment?** | | | | | |
| Yes | 15 (100%) | 4 (80%) | 5 (50%) | 8 (66.7%) | 32 (76.2%) |
| No | 0 (0%) | 1 (20%) | 3 (30%) | 3 (25%) | 7 (16.7%) |
| Not known | 0 (0%) | 0 (0%) | 2 (20%) | 1 (8.3%) | 3 (7.1%) |
| **Total patients with NAT-guided consultation** | **15** | **4** | **5** | **8** | **32** |
| **Clinician present in NAT-guided** | | | | | |
| GP | 6 (40%) | 4 (100%) | 0 (0%) | 8 (100%) | 18 (56.3%) |
| Nurse | 9 (60%) | 0 (0%) | 5 (100%) | 0 (0%) | 14 (43.7%) |
| **NAT-C forms completed in full?** | | | | | |
| Yes | 7 (46.7%) | 3 (75%) | 0 (0%) | 5 (63%) | 15 (47%) |
| No | 8 (53.3%) | 1 (25%) | 5 (100%) | 1 (13%) | 15 (47%) |
| Missing–data not available | 0 | 0 | 0 | 2 (25%) | 2 (6%) |

*(Continued)*

**Table 2.** (Continued)

| | Directed encounter | | Undirected encounter | | Total |
|---|---|---|---|---|---|
| | Site 3 | Site 4 | Site 1 | Site 2 | |
| **Length of appointment (minutes)** | | | | | |
| 10–15 | 1 (6.7%) | 0 (0%) | 0 (0%) | 1 (16.7%) | 2 (7.4%) |
| 15–20 | 9 (60%) | 2 (100%) | 1 (25%) | 0 (0%) | 12 (44.4%) |
| 20–25 | 0 (0%) | 0 (0%) | 1 (25%) | 3 (50%) | 4 (14.8%) |
| 25–30 | 1 (6.7%) | 0 (0%) | 1 (25%) | 1 (16.7%) | 3 (11.1%) |
| >30 | 4 (26.7%) | 0 (0%) | 1 (25%) | 1 (16.7%) | 6 (22.2%) |
| Missing | 0 | 2 | 1 | 2 | 5 |

[a] Reason no needs assessment scheduled: One patient withdrew due to illness; One was admitted to hospital shortly after consent and subsequently deceased; one patient forgot to arrange appointment and developed memory problems.

[b] Reason non-attendance: Two DNA (one of which was subsequently withdrawn by study team for loss of capacity to provide consent)

'changes in your sexual relationships' and 'work around the home'. At baseline, 34/47 (72%) participants reported at least one moderate to severe unmet need [Table 3]. This reduced to just over half at one month, remaining at this level throughout follow up. Level of unmet needs between the arms of the study were similar and are presented overall.

A contingency table [S3 File] reporting needs from baseline to all time-points shows little percentage change within 'no need' and 'unmet need' groups. However, change in mean scores were slightly reduced within all domains of the SCNS at three months, the proposed primary end-point of a definitive trial [S4 File].

**Secondary outcome measures.** Seventeen carers completed study questionnaires at baseline and one month, 14 at three months and 10 at six months. Summary PROMs data are reported [S5 File].

**Process evaluation.** *NoMAD Survey.* Nearly all (28/29 [97%]) of NAT-C trained clinicians completed the NoMAD survey [S6 File]. Most (27/28, [96%]) agreed or strongly agreed that they would support the use of the NAT-C, one (4%) was unsure and nobody did not. Interestingly, 96% of clinicians agreed or strongly agreed that feedback about the NAT-C could be used to improve it in the future. No clinician disagreed or strongly disagreed that they could modify their practice using the NAT-C. The primary concern of clinicians was about resources for delivery, and all clinicians agreed or were unsure that key people may be needed to drive the NAT-C forward.

*Interviews and focus groups.* Four focus groups (n = 11) and 15 patient-participant interviews, three with their carers (six from arm 1; nine from arm 2) were conducted. A single theme was developed which informs our feasibility outcomes: Information and Research.

## Information and research

Clinician, patient and carer-participants supported the need for a definitive trial. Patient-participants expressed altruism regarding trial participation and did not expect personal benefit. Study questionnaires were acceptable, understandable and helpful to patient and carer-participants–despite some repetition. Some participants expressed receiving therapeutic benefit from their completion and/or promotion of healthcare-seeking behaviour. Some patient-participants were confused about whether questionnaire responses were with respect to their cancer, or their health issues overall.

**Table 3. Supportive Care Needs Survey: Descriptive statistics.**

| SCNS Domain | Baseline N = 47 | 1 month N = 44 | 3 months N = 38 | 6 months N = 32 |
|---|---|---|---|---|
| *Psychological*[b] | | | | |
| *No to low need* | 24 (51%) | 30 (68%) | 22 (61%) | 18 (56%) |
| *Unmet need* | 23 (49%) | 14 (32%) | 14 (39%) | 14 (44%) |
| *Mean (SD)* | 23.1 (10.44) | 21. 1 (10.21) | 20.9 (10.50) | 22.4 (10.20) |
| *95% confidence intervals* | 20.1, 26.1 | 21.1, 27.1 | 17.5, 24.3 | 18.9, 25.9 |
| *Median (Range)* | 21 (10,50) | 19.5 (10–44) | 19 (10–44) | 20 (10–43) |
| *Missing items*[a] | 0 | 0 | 2 | 0 |
| *Health system and information*[b] | | | | |
| *No to low need* | 30 (64%) | 34 (79%) | 28 (74%) | 27 (84%) |
| *Unmet need* | 17 (36%) | 9 (21%) | 10 (26%) | 5 (16%) |
| *Mean (SD)* | 25.3 (11.40) | 21.2 (11.74) | 20.3 (10.38) | 19.13(11.70) |
| *95% confidence intervals* | 22.0, 28.6 | 17.7, 24.7 | 17.0, 23.6 | 15.1, 23.2 |
| *Median (Range)* | 22 (11,55) | 21 (11–55) | 20 (11–55) | 15 (11–55) |
| *Missing items*[a] | 0 | 1 | 0 | 0 |
| *Physical and daily living*[b] | | | | |
| *No to low need* | 21 (45%) | 29 (64%) | 21 (58%) | 19 (59%) |
| *Unmet need* | 26 (55%) | 15 (34%) | 15 (42%) | 13 (41%) |
| *Mean (SD)* | 12.4 (4.97) | 10.9 (5.63) | 10.8 (5.42) | 11.3 (4.99) |
| *95% confidence intervals* | 10.9, 13.8 | 9.2, 12.6 | 9.0, 12.6 | 9.6, 13.0 |
| *Median (Range)* | 12 (5,25) | 10 (5–22) | 9 (5,23) | 11 (5–20) |
| *Missing items*[a] | 2 | 0 | 2 | 0 |
| *Patient care and support*[b] | | | | |
| *No to low need* | 35 (74%) | 34 (79%) | 31(86%) | 28 (88%) |
| *Unmet need* | 12 (15%) | 9 (21%) | 5 (14%) | 4 13%) |
| *Mean (SD)* | 9.8 (3.9) | 9.3 (4.73) | 8.1 (4.22) | 7.8 (4.05) |
| *95% confidence intervals* | 8.7, 10.9 | 7.9, 10.7 | 6.7, 9.5 | 6.4, 9.2 |
| *Median (Range)* | 9 (5,21) | 8 (5–22) | 6 (5–19) | 6 (5–20) |
| *Missing items*[a] | 0 | 1 | 2 | 0 |
| *Sexuality*[b] | | | | |
| *No to low need* | 39 (83%) | 37 (88%) | 32 (94%) | 28 (93%) |
| *Unmet need* | 8 (17%) | 5 (12%) | 2 (6%) | 2 (7%) |
| *Mean (SD)* | 5.6 (3.15) | 4.7 (2.60) | 4.8 (2.87) | 4.6 (2.63) |
| *95% confidence intervals* | 4.7, 6.5 | 3.9, 5.4 | 3.8, 5.8 | 3.7, 5.5 |
| *Median (Range)* | 5 (3,15) | 3 (3–13) | 3.5 (3,15) | 3 (3–13) |
| *Missing items*[a] | 0 | 2 | 4 | 2 |
| *Total needs*[b] | | | | |
| *No to low need* | 13 (28%) | 26 (59%) | 19 (50%) | 16 (50%) |
| *Unmet need* | 34 (72%) | 18 (41%) | 19 (50%) | 16 (50%) |
| *Missing domain items*[c] | 0 | 0 | 0 | 0 |
| *Mean (SD)* | 76.1 (26.76) | 67.5 (31.16) | 65.8 (30.04) | 64.9 (29.89) |
| *95% confidence intervals* | 68.2, 83.9 | 58.1, 76.9 | 55.3, 76.2 | 54.2, 75.6 |
| *Median (Range)* | 69 (35,154) | 61.5 (34,140) | 56 (34,144) | 55 (34,139) |
| *Missing items* | 2 | 2 | 4 | 2 |

[a] Missing relates to patient-participants with scores missing due to >50% missing items on the scale where imputation was not possible. Missing items were further present across time points for up to three participants on any one domain, where imputation was possible.

[b] Higher mean scores represent greater unmet need. The range of scores across domains are: 3–15 sexual needs, 5–25 physical needs, 10–50 psychological needs, 11–55 health system and information needs, 5–25 patient care and support needs.

[c] Patient-participants were included in 'total needs' if they had complete data for ≥50% of individual domains.

*Illustrative quotes.* "[The NAT-C] was more like a guide, if anything, because you don't, the skill is trying to sit there with the tool and make sure you address each of the items in the normal consultation. So you don't want to look, make it look like an interview because then you lose that rapport, don't you, with the patient?" [Clinician 1a]

"I will say though that having spoken to [GP name] this morning I will, I will use the GP more, it was more, not necessarily accessible, cos that's an issue I do have, accessing the GP, but it felt more personal this morning, definitely." [Patient 46]

"If I see a patient who I know pretty well who's had, or who has active cancer, I use the tool at that point, just to check in really. . . the fact that there are so many places along the patient journey where information gathering could be missed it's better really to use the tool; that way you're picking up quite a lot." [Clinician 2a]

Clinicians were unsure of the effectiveness of tools currently available. Some saw the potential importance of the NAT-C and as a way of conducting a cancer review but wanted an evidence-base for its use. Many were concerned about the time needed, fearing longer than the allotted 20 minutes. Most clinicians used a paper NAT-C despite an available electronic version, albeit limited in its functionality. They suggested a fully embedded electronic version would be useful. Doctors used the tool as an *aide memoire* rather than a series of questions, consistent with the tool design but nurses used the NAT-C as a questionnaire. Some GPs reported continued NAT-C use post-trial.

The process evaluation identified early recruitment issues and informed protocol amendments (inclusion of patients with active cancer, alteration of study approach). Avoidable time-delays in trial set-up included local delays for sponsor approval. This trial was also set-up during changes in research governance, with increased responsibility for the HRA. This led to confusion regarding when study sites could be approached in relation to ethics approval. Further avoidable delays followed ethical approval due to availability for study training.

## Stop and go criteria

Overall, trial results were favourable in relation to the pre-defined stop/go criteria and proposed modifications to study design for Phase III study presented [Table 4].

**Changes in definitive trial design in response to feasibility findings.** The definitive trial will be a two-armed parallel group Cluster Randomised Controlled Trial (cRCT) with internal pilot, embedded process evaluation and cost effectiveness evaluation. Although recruitment was higher in the undirected encounter arm, directed encounter ensured more NAT-C guided appointments and more acceptable to GP practices, who all requested direct arrangement of appointments. Participants raised no concerns with being directed towards a specific clinician.

*Primary outcome.* Unmet need identified in any domain in the Supportive Care Needs Survey SF34 (SCNS) at three months post registration. Three months was chosen as the primary endpoint to allow effects from referrals to manifest and due to high retention of participants at three months during feasibility (81%). Six months was considered too distant from the intervention to directly link any effect, however, differences between arms at six months will be a secondary outcome.

*Secondary outcomes.* Secondary outcome measures are reported in S1 Box. Participants in the feasibility study reported positive experiences of completing the study questionnaires apart from the RUQ, where some experienced difficulty. We have therefore developed a revised version of the questionnaire for the definitive study. The CSNAT will also be replaced by the Zarit-6 to assess carer burden, due to concerns amongst the Trial Management Group relating to the similarity between the CSNAT and the NAT-C.

**Table 4. Stop and go criteria outcomes.**

| Stop / Go Assessment | Stop / go criteria | | Outcome and proposed modifications to study design for Phase III study |
|---|---|---|---|
| **Recruitment** A minimum of 10–15 patients should be recruited from each general practice over 6 months to demonstrate an acceptable recruitment rate to progress to the cRT | Red: | <7 patients recruited -Insufficient numbers per practice to proceed | Site 1: 12 patients recruited (green) |
| | | | Site 2: 14 patients recruited (green) |
| | | | Site 3: 15 recruited (green) |
| | | | Site 4: 6 patients recruited (red) |
| | Amber: | 7–10 patients recruited—Sufficient numbers per practice to proceed with changes | Overall: On average, 12 patients recruited per site |
| | | | Green–continue to Phase III trial |
| | Green: | 10–15 patients recruited—continue to Phase III trial | **Proposed modifications for Phase III study** • Modified study approach, to include Expression of Interest forms • Allow research nurse follow up to patient invitations |
| **Uptake and delivery** Attendance at the initial NAT-C GP appointment within one month post registration (baseline/consent): | Red: | <50% NAT-C appointments within one month–insufficient numbers to proceed | **Amber:** 76% of patient participants received a study appointment within one month of registration, 68% received a NAT-C guided appointment. |
| | Amber: | 50–80% NAT-C appointments within one month–sufficient numbers to proceed with changes | Continue to Phase III trial with modifications to study processes. |
| | Green: | >80% NAT-C appointments within one month–continue to Phase III trial | **Proposed modifications for Phase III study** • Patient participants to be directed towards a known NAT-C trained clinician • GP practices to proactively arrange appointments with patients |
| **Data collection and quality** Proposed primary outcome measure, the SCNS, follow up completion rate at 3 months (completed: face to face, via telephone or postal return) | Red: | <65% SCNS completion at three months—insufficient numbers to proceed | **Green:** SCNS completion rate at 3 months: 81% |
| | | | Continue to Phase III trial |
| | Amber: | 65–80% SCNS completion at three months—sufficient numbers to proceed with changes | **Proposed modifications for Phase III study** • A trial research nurse, not trial academic, to conduct follow up phone calls |
| | Green: | >80% SCNS completion at three months–continue to Phase III trial | |

*Changes to the intervention.* Reflecting clinician views identified during process evaluation that the NAT-C could be improved based upon feedback, we have developed templates for the NAT-C in EMIS and SystmOne to address clinician preference that the NAT-C be integrated in to existing computer systems.

**Sample size.** For the definitive trial, we estimate that a sample size of 1080 patients from 54 GP practices is needed to detect a 22% relative improvement in proportion of patients reporting unmet on the SCNS-SF34 (85% power, 5% significance level, ICC 0.05, 20% loss to follow-up). During feasibility, sites recruited on average two patients per month over six months. To recruit our sample size, we will allow a recruitment phase of 21 months, assuming one patient per practice per month.

**Randomisation.** Overall, recruitment rates during feasibility achieved the stop/go criteria. However, recruitment varied between practices, one site recruited 15 patients, whilst another with a larger practice list size, recruited only six.

Variation in recruitment between sites is to be expected, but not contingent upon general practice list size. To account for potential variability between sites and to ensure that findings are representative, randomisation for the definitive study will be stratified by: locality (urban or rural location), GP list size GP list size: Small (<5000), medium (5000–10000) and large (>10000)) and whether sites serve as training practices (yes/no).

*Patient inclusion criteria.* We will include all patients with active cancer, regardless of intention of treatment from the outset. Identifying patients undergoing treatment with palliative

intent was challenging, often due to a lack of clarity *regarding* treatment intent in Oncology clinic letters on GP systems during screening. Including all patients with active cancer is further warranted, based upon high levels of unmet need at baseline (72%) amongst participants.

**Process evaluation.** During feasibility, the primary focus of process evaluation was to investigate feasibility outcomes and issues potentially limiting successful completion of the trial and identify solutions. For the definitive trial, process evaluation will use NPT to seek to understand whether the NAT-C can be implemented and embedded into practice, and if so what approach to implementation would work best. By examining implementation alongside clinical effectiveness evaluation, we will be able provide an answer not merely to the question 'Should we implement NAT-C into practice?' but also 'How should we implement NAT-C into practice?'

**Changes to study processes.** *Study set-up*. Process evaluation identified delays associated with regulatory approvals (HRA, local Sponsor). Our feasibility trial took place during the embedding of new processes of HRA approval, which we are informed are resolved. Further delays were caused by logistical issues relating to booking appointments to conduct site set-up and clinician training following ethical approval. In the definitive trial we will arrange site-initiation appointments in anticipation of ethical approval.

**Participant recruitment.** Early problems with slow recruitment were resolved by including an EOI with study invitation and allowing research nurses to conduct follow-up phone calls. Patients at all cancer stages valued the opportunity to take part in research and the NAT-C was considered appropriate. In a definitive trial, we will include EOI forms within the study approach and gain approval for practice staff to conduct follow up phone calls. Participants valued the opportunity to take part in research, supported a definitive trial and incurred no harms.

**Uptake and delivery.** More GPs were trained than nurses but this varied across, potentially reflecting GP availability issues. Reduction of time from training to study appointments, improved recruitment approach and availability of re-training opportunities (online training, webinars) are planned to minimise NAT-C trained clinicians forgetting to use the NAT-C. Median time of 15–20 minutes per NAT-C appointment allays prior concerns about implementation into routine practice–including as a cancer review—consistent with our previous study [18]. Participating practises will nominate key clinicians to be trained and deliver the NAT-C in a definitive trial. Face to face clinician training promoted useful discussion between clinicians regarding how to use the NAT-C. To account for any influence upon the delivery of the intervention, question and answer sessions will take place at the end of face-to-face training and webinars and some frequently asked questions will be available as part of online training.

**Data collection and quality.** Patients and carers had confidence in study questionnaires and some felt they were beneficial and/or had changed health-seeking behaviour. When used in a joint patient-clinician consultation, PROMS can change how a patient feels about their condition and influence the relationship with clinician [19, 20]. However, this effect is uncertain when used as an isolated measure, rather than as an integral part of joint clinical decision-making [21]. Thus baseline measures in the definitive trial will include PROMs to maximise study power.

## Discussion

### Informing a definitive trial and lessons for potential implementation of the NAT-C

Our findings suggest that with the alterations to trial design made during feasibility work, that a multi-centre cluster Randomised Controlled Trial (cRCT) to test the effectiveness of the NAT-C compared to usual care is feasible. Early recruitment problems were resolved by broadening our inclusion criteria from patients 'with incurable cancer', to patients with 'active cancer' and the addition of an EOI form with study invitations. Our patient sample included

9/47 (19%) of participants with metastatic cancer. This suggests that our patient identification methods were appropriate and that people with terminal cancer were willing to take part and that it was the alteration of the study approach which improved recruitment. Data demonstrate a high prevalence of unmet patient needs at baseline. This study was not designed to demonstrate effectiveness of the intervention, but the high prevalence, uptake of the NAT-C, successful follow-up and completion of outcomes, and patient interviews support the need for a definitive trial. Findings from our process evaluation that 89% of clinicians see the importance of the NAT-C for their work support the suggestion that primary care has a major role to play in palliative care [22]. Challenges during feasibility were consistent with other primary care clinical trials; 56% with time extensions and 18% with altered recruitment methods [23]. Use of multiple methods and process evaluation has allowed us to use data-driven solutions, to refine the study design of a definitive trial, to maximise prospects of delivery of the study.

We will conduct a cRCT to test the clinical effectiveness and cost-effectiveness of the Needs Assessment Tool—Cancer (NAT-C) in primary care for people with active cancer with regard to unmet patient and caregiver need, compared with usual care. Feasibility and process evaluation findings have informed several changes to the original design of the feasibility trial.

By broadening the inclusion criteria to all those with active cancer, if effective, the NAT-C could be a validated and tested way of conducting cancer care reviews. It may also inform a consistent, evidence-based approach to referring cancer patients for specialist palliative care input [24]. Referral would be needs-based rather than prognosis-based facilitating earlier integration of palliative care and improve clinical outcomes [25, 26].

Feasibility work is an essential part of trial preparation, in particular for trials of complex interventions [27]. Our mixed-methods feasibility trial has informed the design of a definitive trial to maximise the likelihood of successfully conducting a definitive cRCT, to test the effectiveness of the NAT-C in reducing the unmet needs of cancer patients and their carers, compared with usual care.

Finally, feasibility work took place prior to the COVID-19 outbreak in the UK. COVID-19 has brought changes in how primary care is delivered for cancer patients, many of whom have been shielding, with remote consultations becoming commonplace and protective equipment necessary for face-to-face appointment. Concerns that cancer patients may have received a reduced level of care during this time may have further increased unmet needs and increased the justification for the study [28].

We have therefore amended study processes and interventions to allow remote delivery for a definitive trial, including online and audio-recorded consent, if necessary, given the ongoing cases and the risk of a second wave.

## Strengths and limitations

A strength of this study is the use of mixed-methods to assess feasibility. Emerging findings from the NPT study enabled solutions to problems to be identified and implemented successfully during the feasibility trial. In particular, the combination of quantitative and qualitative data were informative regarding choice of primary and secondary outcome measures, and mode of delivery of the NAT-C (directed rather than un-directed).

There were a number of weaknesses. Firstly, consenting patients for interview those who had already expressed support for the study through their participation, provides an inherent bias in the interview study. Of the two who withdrew for personal reasons, neither wished to be interviewed. Patients dissatisfied with their cancer care may be less likely to participate. Secondly, our patient/carer sample was entirely White British, unrepresentative of the general population. Lastly, our randomisation tested method of intervention invitation, but not

willingness to be randomised or consent materials. We will therefore test this aspect in the nested pilot study of the definitive cRCT.

## Conclusion

Our feasibility study identified a high proportion of people with cancer with unmet supportive and palliative care needs. This supports the rationale for a definitive study of the NAT-C, a needs assessment tool which encourages systematic identification of patient problems and prompts action. Cancer patients and clinicians support the rationale for the trial and were willing to undergo study measures and deliver the NAT-C respectively. Our findings suggest that a definitive trial of the NAT-C in primary care is feasible, with appropriate refinement, in terms of GP practice and patient participant recruitment, data quality and acceptability and use of the NAT-C.

## Supporting information

**S1 File. Illustrative questions from interview guides: GP practice staff, patients and carers.**
(DOCX)

**S2 File. Baseline characteristics of patient participants with and without follow-up (missing data).**
(DOCX)

**S3 File. Supportive Care Needs Survey contingency table.**
(DOCX)

**S4 File. Change in mean score compared to baseline on the Supportive Care Needs Survey at each time point.**
(DOCX)

**S5 File. Summary scores of secondary outcome measures.**
(DOCX)

**S6 File. Clinician responses to the NoMAD survey (n = 28).**
(DOCX)

**S7 File.**
(DOCX)

**S8 File.**
(DOC)

**S9 File.**
(DOCX)

**S1 Study.**
(DOCX)

**S1 Carer data.**
(SAV)

**S1 Patient data.**
(SAV)

**S1 Box. Feasibility outcomes, assessment measures and stop/go criteria.**
(DOCX)

## Acknowledgments

We acknowledge the contribution of participating GP practices and in particular, support from Carla Bratten of the Clinical Research Network, Dr Robbie Foy, University of Leeds and Florence Reedy, University of Hull.

## Author Contributions

**Conceptualization:** Elvis Amoakwa, Alexandra Wright-Hughes, John Blenkinsopp, David C. Currow, David Meads, Amanda Farrin, Victoria Allgar, Una Macleod, Miriam Johnson.

**Data curation:** Joseph Clark, John Blenkinsopp, Miriam Johnson.

**Formal analysis:** Joseph Clark, Alexandra Wright-Hughes, John Blenkinsopp, David C. Currow, Victoria Allgar, Miriam Johnson.

**Funding acquisition:** Alexandra Wright-Hughes, David C. Currow, David Meads, Amanda Farrin, Victoria Allgar, Una Macleod, Miriam Johnson.

**Investigation:** Elvis Amoakwa, David C. Currow, David Meads, Amanda Farrin, Miriam Johnson.

**Methodology:** Alexandra Wright-Hughes, John Blenkinsopp, David C. Currow, David Meads, Amanda Farrin, Victoria Allgar, Una Macleod, Miriam Johnson.

**Project administration:** Amanda Farrin, Miriam Johnson.

**Resources:** Miriam Johnson.

**Software:** Miriam Johnson.

**Supervision:** Alexandra Wright-Hughes, John Blenkinsopp, David C. Currow, David Meads, Amanda Farrin, Miriam Johnson.

**Validation:** Miriam Johnson.

**Visualization:** Miriam Johnson.

**Writing – original draft:** Joseph Clark.

**Writing – review & editing:** Joseph Clark, Elvis Amoakwa, Alexandra Wright-Hughes, John Blenkinsopp, David C. Currow, David Meads, Amanda Farrin, Victoria Allgar, Una Macleod, Miriam Johnson.

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
