## [Decision Letter · Decision Letter 0]

18 Nov 2020

PONE-D-20-25507

A cluster randomised trial of a Needs Assessment Tool for Cancer (NAT-C) in primary care: a feasibility study

PLOS ONE

Dear Dr. Clark,

Thank you for submitting your manuscript to PLOS ONE. After careful consideration, we feel that it has merit but does not fully meet PLOS ONE’s publication criteria as it currently stands. Therefore, we invite you to submit a revised version of the manuscript that addresses the points raised during the review process.

The reviewers have made a number of points that could be expanded upon in the manuscript, including several points of clarification related to participant recruitment, sample size, and missing data.

We look forward to receiving your revised manuscript.

Kind regards,

Adam Todd, PhD

Academic Editor

PLOS ONE

Journal Requirements:

a) Did participants provide their written or verbal informed consent to participate in this study?

Reviewers' comments:

Reviewer's Responses to Questions

**Comments to the Author**

1. Is the manuscript technically sound, and do the data support the conclusions?

Reviewer #1: Yes

Reviewer #2: Partly

2. Has the statistical analysis been performed appropriately and rigorously? 

Reviewer #1: Yes

Reviewer #2: N/A

3. Have the authors made all data underlying the findings in their manuscript fully available?

Reviewer #1: Yes

Reviewer #2: Yes

4. Is the manuscript presented in an intelligible fashion and written in standard English?

Reviewer #1: Yes

Reviewer #2: Yes

5. Review Comments to the Author

Reviewer #1: The primary objective of this cluster randomized trial is to assess the feasibility and the acceptability of the NAT-C in primary care by evaluating the recruitment of GP practices, patients and carers; delivery/uptake of the NAT-C, follow-up, acceptability of study measures and data quality. This is an interesting topic, however, there are some experimental design and statistical data analysis concerns about this paper.

Critiques:

1. The authors stated that “After slow initial recruitment, inclusion criteria were amended to include patients with active cancer irrespective of treatment intent. A further amendment allowed inclusion of a pre-paid Expression of Interest Form (EOI) with study invitations and patient information.” Please provide the quantitative measurements of the recruitment rate before and after each amendment and comment on the rationale of these amendments.

2. The sample size justification is superficial. The authors should provide the quantitative adjustments, e.g., the precision analysis, in addition to a reference.

3. The authors stated that “the mean response was used to impute missing item(s)” (line 171). This is not recommended because the variation is underestimated. Please use the multiple imputation approach for missing data analysis.

4. In addition to reporting the mean and SD, the authors should also report the 95% confidence interval for the outcome measurements.

5. The authors should condense the article to improve the readability.

Reviewer #2: This paper reports on a feasibility study of an intervention incorporating the NAT-C tool that targeted patients with cancer and was undertaken in preparation for a definitive trial evaluation. It’s an interesting topic. However, I found the manuscript difficult to follow in parts. As per comments below, I think more background information is needed as well as a clearer overview of what the intervention entailed. I would encourage the authors to also consider the use of a reporting guideline for reporting randomised pilot and feasibility studies conducted for the overall paper (https://www.equator-network.org/reporting-guidelines/consort-2010-statement-extension-to-randomised-pilot-and-feasibility-trials/ ) as well as for the intervention (https://www.bmj.com/content/348/bmj.g1687)

Title

I find the reference to two study designs in the title confusing

There is no mention of palliative care in the title even though the abstract explicitly mentions palliative care

Abstract: this would benefit from further review/refinement. I appreciate the challenge in writing an abstract but some of the methods subsection doesn’t give much insight. The following sentences are particularly vague: “Study questionnaires at baseline and 1, 3 and 6 months. Participants booked a 20 minute study appointment post-baseline. Patients, carers and GP practice staff were invited to interviews/focus groups.” For example, there is no indication of what the study questionnaire or post baseline appointment involved. The target patient population needs to be more clearly specified ; “patients with active cancer” is not terribly informative (it doesnt event differentiate between adults and children). There is no clear sense of what the intervention was from the abstract

Wording of NAT-C differs from title above

Results subsection: “Fifteen patient interviews and focus group at each GP practice.” – not a complete sentence and not sure what this is adding

“Feasibility outcomes informed the design of a 2-armed cluster randomised controlled trial to test the effectiveness and cost-effectiveness of the NAT-C compared with usual care.” Is this a direct result of the study? It sounds more like the next step. Feasibility outcome have not been outlined in the methods subsection

Conclusion: I don’t think the relevant results have been outlined to support this; there is also no mention of refinements until this last sentence which somewhat jars with the previous statement that participants found measures acceptable

Introduction

General comments: The introduction doesn’t adequately set the scene; the opening two paragraphs don’t provide enough relevant background and instead most of it discusses plans for a future trial. The structure of the introduction could also benefit from refinement; it should flow into a final paragraph that outlines the aim and objectives but at present these seem to be split across the third and fourth paragraphs; the outline of objectives in both of these paragraphs is confusing. I would encourage the authors to consider the Medical Research Council’s framework on the development/evaluation of complex interventions and think about describing the intervention (including how it has been developed) and then working in the need for preliminary testing before a more definitive evaluation.

Specific comments

A stronger opening paragraph is needed. The opening sentence is particularly vague – unmet needs could refer to anything. This needs to set the scene in terms of the scale of the problem that needs to be addressed so that you can build up to outlining specific gaps that this research is addressing later in the introduction

“triage tailored action and audit and service planning” what does this mean/entail

From paragraph 2, I have no real sense of what the tool is or consists of. From a clinical perspective, the name of the tool is quite non-specific (apart from referring to cancer) and does not give any clear indication that it might link to palliative care

Are there any references to back up the points about the tool made in paragraph 3?

This is the first time “supportive and palliative care needs” is mentioned; the target population needs to be outlined more clearly and whether this is about providing supportive/palliative care from the point of initial diagnosis or whether it’s more focused on advanced disease and end of life

“by supporting clinicians to systematically assess patient/carer problems across multiple domains” what are these domains that you are referring to

What do you mean by phase 3 in the context of a clinical intervention? See point regarding MRC framework above

Not sure why objectives for a future study are being presented; suggest keeping the focus on the study that is being presented

Paragraph 4;

Objective 1 is actually two objectives – recruitment and training

Objective 2; I don’t understand the link between recruitment rate and method of delivery

Objective 3; I don’t understand this objective

Objective 4; what does uptake mean?

Objective 5; is this more about data collection measures?

There is no clear mention of fidelity even though is reported later in the results

The final sentence regarding stop/go criteria seems more relevant to methods than introduction

Methods

“The Cancer Needs Assessment (CANASSESS) study was a non-blinded, parallel non-controlled, feasibility trial with 82 cluster randomisation to method of delivery of the NAT-C, qualitative sub-study and parallel process evaluation, in four General Practices.” – very long sentence

Why wasn’t a usual care group included? Surely this would be critical to a future trial?

Participants and setting; more information needed about where practices were based; how they were identified, contacted, recruited etc

“Randomisation was with regard to the method of delivery of the NAT-C” this is the first mention of this being able to be delivered in different ways. Goes back to previous comments about intro not setting the scene adequately

“test uncertainty on two counts” what does this mean??

“Eligible patients were consenting adults with incurable cancer” first clear mention that the focus is on terminal patients; this makes me query the rationale for focusing on general practice – would these patients not be linked to palliative care services?? This should be teased out in the introduction

“After slow initial recruitment” how long had the study been underway when this change was implemented? This is an interesting feasibility finding and links back to my previous comment above

I would also like to see dates worked in somewhere within the methods of when the study formally started and finished

The intervention is core to this study yet it is quite poorly described. I have no real sense of what issues were focused on and how GPs were expected to address them – particularly in cases where issues/challenges might have been outsider of their scope of practice etc. There is so much emphasis now on providing sufficient information to enable replication – please consider the use of a reporting guideline such as TiDiER

Study assessments and outcomes; abbreviations not defined when first used – same with NoMAD; they would also benefit from some elaboration

“Feasibility outcomes, assessment measures and stop/go criteria are presented in Box 1.” I’d like to see an outline of key feasibility outcomes listed in the main text. The primary outcome and the tool being used to assess it needs to be outlined in the main text – particularly in terms of choice and suitability

Did participants receive any incentives?

Ethics approval and consent to participate: Unless there is specific journal requirements around this content; I think this could be condensed

Results

I think the dates would be better placed in methods

Overall this section is long and a little hard to follow in places – I’d like to see the reported findings more clearly linked to the study objectives/feasibility questions. At present there are just too many subheadings for the reader to navigate. There may also be opportunity for condensing some the information which would enable the introduction and discussion to be teased out more

“Three main themes were generated. Here, we present the theme which informs our feasibility outcomes: Information and Research.” I don’t follow this

I don’t know why quotes are presented with their own separate headings

There seems to be very little focus on the patient interview findings in the results presented

Discussion

A stronger and more comprehensive discussion is needed that places findings in context of relevant literature. Structure/flow needs refining; no opening paragraph, just a single sentence and then focuses on future work; focus on discussing the work presented and build up to how the findings support you progressing to the next stage of evaluation. There are other cases where single sentences are presented instead of complete paragraphs

I would also contest your opening statement as you didn’t succeed in recruiting the specific group of patients that you had originally set out to recruit

“The high prevalence, and that we cannot apportion causality to the intervention in this non

controlled trial supports a definitive trial.” I don’t know if I agree with the first point regarding prevalence – the study wasn’t designed to assess this; I don’t understand the logic/rationale behind the second point regarding causality

“Primary care has a major role to play in palliative care [19].” Again goes back to issues flagged with introduction and no detailed explanation of rationale for focusing on general practice in this study; also this isn’t a very informative statement

“Challenges during feasibility were consistent with other primary care clinical trials; 56% with time extensions and 18% with altered recruitment methods [20]. Use of multiple methods and process evaluation has allowed us to use data driven solutions, to refine the study design of a definitive trial, to maximise prospects of delivery of the study.” I’d like to see more specific discussion of your study findings and them then being placed in context of the literature

“GP practices receive Quality and Outcomes Framework payments for conducting cancer reviews within six months of patients’ diagnosis. However, there is no guidance regarding what a cancer review should involve or evidence-based template available.” This needs to be introduced much much earlier in the paper!

“By broadening the inclusion criteria to all those with active cancer, if effective, the NAT-C could be a validated and tested way of conducting cancer care reviews. It may also inform a consistent, evidence-based approach to referring cancer patients for specialist palliative care input [21]. Referral would be needs-based rather than prognosis-based facilitating earlier integration of palliative care and improve clinical outcomes [22, 23].” - I’d be interest in the authors' view about why they were unable to recruit the intended patient population and what could be done about this? it seems to me that there is a risk that by changing the patient inclusion criteria the intervention still doesn’t reach the original target group (i.e. those with a terminal diagnosis )

“Feasibility work is an essential part of trial preparation, in particular for trials of complex interventions [24].” Again; more relevant to setting the scene in introduction not the tail end of the discussion

First mention of mixed methods is in the discussion

Conclusion

A more cohesive/comprehensive concluding paragraph is needed; I don’t think that you can say in advance that a definitive trial is feasible (you’ll only find that out once you get it underway) - need to focus on the feasibility questions that this study has answered and link that to whether it supports progressing to a definitive evaluation

6. PLOS authors have the option to publish the peer review history of their article (what does this mean?). If published, this will include your full peer review and any attached files.

Reviewer #1: No

Reviewer #2: No

---

## [Author Response · Author response to Decision Letter 0]

9 Dec 2020

Thank you, we have uploaded a Word Document which details our responses to each point raised by reviewers and editors and also highlighted some key changes within our revised letter to the editor.

---

## [Decision Letter · Decision Letter 1]

6 Jan 2021

A cluster randomised trial of a Needs Assessment Tool for adult Cancer patients and their carers (NAT-C) in primary care: a feasibility study

PONE-D-20-25507R1

Dear Dr. Clark,

We’re pleased to inform you that your manuscript has been judged scientifically suitable for publication and will be formally accepted for publication once it meets all outstanding technical requirements.

Kind regards,

Adam Todd, PhD

Academic Editor

PLOS ONE

Additional Editor Comments (optional):

Reviewers' comments:

Reviewer's Responses to Questions

**Comments to the Author**

1. If the authors have adequately addressed your comments raised in a previous round of review and you feel that this manuscript is now acceptable for publication, you may indicate that here to bypass the “Comments to the Author” section, enter your conflict of interest statement in the “Confidential to Editor” section, and submit your "Accept" recommendation.

Reviewer #1: All comments have been addressed

Reviewer #2: All comments have been addressed

2. Is the manuscript technically sound, and do the data support the conclusions?

Reviewer #1: Yes

Reviewer #2: Yes

3. Has the statistical analysis been performed appropriately and rigorously? 

Reviewer #1: Yes

Reviewer #2: N/A

4. Have the authors made all data underlying the findings in their manuscript fully available?

Reviewer #1: Yes

Reviewer #2: Yes

5. Is the manuscript presented in an intelligible fashion and written in standard English?

Reviewer #1: Yes

Reviewer #2: No

6. Review Comments to the Author

Reviewer #1: The authors have responded well to the statistical issues raised in the previous review. There is no further statistical concern about this revised manuscript.

Reviewer #2: Thank you for your detailed responses to my previous comments. The manuscript reads clearer now. I think syntax may need to be reviewed in places throughout

For example, abstract needs additional refinement. Syntax is poor in places

E.g.

“Practices cluster randomised (1:1) to Arm I: promotion and use of NAT-C with a NAT-C trained clinician or Arm II:clinician of choice irrespective of training status.”

“Patients, carers and GP practice staff views regarding the study sought through interviews/focus groups”

“Fortyseven participants and 17 carers recruited”

The word appropriate in the conclusion doesn’t align with the aim

Objectives; wording/syntax needs to be reviewed

e.g. “proportion of patients unmet patient needs”

7. PLOS authors have the option to publish the peer review history of their article (what does this mean?). If published, this will include your full peer review and any attached files.

Reviewer #1: No

Reviewer #2: No

---

## [Editor Report · Acceptance letter]

13 Jan 2021

PONE-D-20-25507R1 

A cluster randomised trial of a Needs Assessment Tool for adult Cancer patients and their carers (NAT-C) in primary care: a feasibility study 

Dear Dr. Clark:

I'm pleased to inform you that your manuscript has been deemed suitable for publication in PLOS ONE. Congratulations! Your manuscript is now with our production department. 

Kind regards, 

on behalf of

Dr. Adam Todd 

Academic Editor

PLOS ONE